# Existence of Circulating Mitochondria in Human and Animal Peripheral Blood

**DOI:** 10.3390/ijms21062122

**Published:** 2020-03-19

**Authors:** Xiang Song, Wei Hu, Haibo Yu, Honglan Wang, Yelu Zhao, Robert Korngold, Yong Zhao

**Affiliations:** Center for Discovery and Innovation, Hackensack Meridian Health, Nutley, NJ 07110, USA; Xiang.Song@HMH-CDI.org (X.S.); whu2@stevens.edu (W.H.); Haibo.Yu@HMH-CDI.org (H.Y.); Honglanwang@yahoo.com (H.W.); yeluzh@gmail.com (Y.Z.); Robert.Korngold@HMH-CDI.org (R.K.)

**Keywords:** Mitochondria, Blood, Plasma, Serum, Immune cells

## Abstract

Mitochondria are usually located in the cytoplasm of cells where they generate adenosine triphosphate (ATP) to empower cellular functions. However, we found circulating mitochondria in human and animal blood. Electron microscopy confirmed the presence of mitochondria in adult human blood plasma. Flow cytometry analyses demonstrated that circulating mitochondria from the plasma of human cord blood and adult peripheral blood displayed the immune tolerance-associated membrane molecules such as CD270 and PD-L1 (programmed cell death-ligand 1). Similar data were obtained from fetal bovine serum (FBS) and horse serum of different vendors. Mitochondria remained detectable even after 56 °C heat inactivation. A real-time PCR array revealed purified mitochondria from animal sera expressed several genes that contribute to human T- and B-cell activation. Transwell experiments confirmed the migration capability of mitochondria through their expression of the chemokine receptor CXCR4 in responses to its ligand stromal-derived factor-1α (SDF-1α). Functional analysis established that human plasma mitochondria stimulated the proliferation of anti-CD3/CD28 bead-activated PBMC, up-regulated the percentage of activated CD4^+^ T and CD8^+^ T cells, and reduced the production of inflammatory cytokines. These findings suggested that the existence of circulating mitochondria in blood may function as a novel mediator for cell-cell communications and maintenance of homeostasis. Plasma-related products should be cautiously utilized in cell cultures due to the mitochondrial contamination.

## 1. Introduction

For decades, mitochondria have been considered as “a cellular power plant” to generate ATP through oxidative phosphorylation and capable of energizing cellular activities. Dysfunction of mitochondria can result in cell apoptosis/death or various cellular stresses including elevated reactive oxygen species (ROS) production, which can contribute to metabolic syndrome, diabetes, autoimmune diseases, and cancer. To date, increasing evidence has revealed the diverse functions of mitochondria in maintaining cell hemostasis through dynamic mitochondrial fusion/fission, metabolic control, interactions with other organelles and the modulation of the nuclear genome [1,2]. Recently, the horizontal transfer of mitochondria have been reported between adjacent cells through the formation of tunneling nanotubes (TNT), cell fusion, GAP junctions, and microvesicles [3]. Under pathological conditions, free or microvesicle-associated mitochondria could be released by the activated monocytes [4], diseased organ cells [5], and oxidative-stressed mesenchymal stem cells [6]. However, it is unknown whether mitochondria are physiologically presented in the blood. 

We previously identified multipotent stem cells from human cord blood (designated cord blood-derived multipotent stem cells, CB-SC) [7] that phenotypically exhibit embryonic transcription factors (e.g., OCT3/4 and SOX2) that distinguish them from mesenchymal stem cell (MSC) and hematopoietic stem cells (HSC) [8]. Based on the comprehensive immune modulation features of CB-SC [9,10], we developed a “stem cell educator (SCE) therapy” that uses allogeneic CB-SC to “educate” a patient’s immune cells, affecting the progression of autoimmunity in subjects with type 1 diabetes (T1D) [9] and other autoimmune diseases such as alopecia areata [11]. SCE therapy provided lasting improvement of pancreatic islet β-cell function and C-peptide (a by-product of insulin production) levels [9]. While exploring clinical mechanisms underlying the long-lasting therapeutic effects of SCE therapy [12] in ex vivo studies, we found that mitochondria released from platelets can migrate to pancreatic islets and be taken up by human islet β cells, leading to the improvement of islet β-cell function and C-peptide production [12], supporting the clinical outcomes of the improved health status in patients with type 1 [9,13] and type 2 diabetes [14]. Platelets along with erythrocytes (red blood cells, RBC) are the largest components of blood and whereas platelets have functional mitochondria, mature RBC do not have them [15]. Therefore, we hypothesized that there were platelet-released mitochondria circulating in the blood. Here, we demonstrate the presence of mitochondria in human and animal blood by flow cytometry and electronic microscopy.

## 2. Results

### 2.1. Characterization of Cord Blood (CB)- and Adult Peripheral Blood (PB) Plasma-Derived Mitochondria

Initially, plasma mitochondria were purified from fresh plasma of cord blood and adult peripheral blood by using the optimized protocol with a serial centrifugation and filtration (Figure 1A). Using human platelets’ mitochondrial staining as positive control, flow cytometry established that MitoTracker Deep Red-labeled CB- and PB-derived mitochondria expressed the mitochondrion-specific marker HSP60 (Figure 1B). The percentages of MitoTracker Deep Red^+^HSP60^+^ mitochondria were 80.11% ± 10.16% for CB-derived mitochondria (*N* = 4) and 74.73% ± 5.06% for PB-derived mitochondria (*N* = 4), relative to the positive control from purified platelet-derived mitochondria (81.85% ± 15.72%, *N* = 4). Additional phenotypic characterization confirmed that these purified, CB- and PB-plasma-derived mitochondria displayed immune tolerance-associated membrane molecules such as CD270 (HVEM, Herpesvirus entry mediator), with 85.76% ± 13.22% and 79.37% ± 8.18% of MitoTracker Deep Red^+^CD270^+^ mitochondria for CB- (*N* = 8) and PB-derived mitochondria (*N* = 9) respectively, and CD274 (programmed cell death-ligand 1, PD-L1), with 81.32% ± 7.39% and 64.33% ± 10.45% of MitoTracker Deep Red^+^CD274^+^ mitochondria for CB- (*N* = 8) and PB-derived mitochondria (*N* = 9) respectively (Figure 1C), as previously characterized on platelet-derived mitochondria (Figure 1C, top panels) [12]. Electron microscopy confirmed the presence of mitochondria in adult blood plasma and showed that the double membranes and mitochondrial cristae were intact with oval (Figure 1D) or elongated morphology (Figure 1E). These data proved the existence of mitochondria circulating human peripheral blood. 

### 2.2. Characterization of Mitochondria from Bovine Serum and Horse Serum

To determine the presence of mitochondria in animal serum, we then examined fetal bovine serum (FBS) and horse serum from different vendors labeled with mitochondrial markers MitoTracker Deep Red or anti-HSP60 mAb and found similar results (Figure 2A). Electron microscopy further confirmed the existence of mitochondria in FBS (Figure 2B,C), suggesting that naturally circulating mitochondria in the blood may be universal in mammals. Importantly, we found that the mitochondria remained detectable with MitoTracker Deep Red even after 56 °C heat inactivation for 30 min (Figure 2D). A real-time PCR array revealed that purified mitochondria from animal sera expressed several genes, which potentially contribute to human T and B cell activation (Figure 2E), even at varied levels from different vendors. Therefore, it implies that existence of mitochondria in FBS may potentially affect the results of all experiments by using FBS as nutrient supplies for their cell cultures.

### 2.3. The Chemotactic Capability of Mitochondria toward a Chemoattractant SDF-1α

To determine chemotaxis potential of plasma mitochondria, we performed the transwell migration assay. Human umbilical vein endothelial cells (HUVEC) were seeded in the top chamber at 1 × 10^5^ cells/mL, in endothelial cell basal medium. The chemoattractant SDF-1α was added into the bottom chamber (Figure 3A). The purified PB-derived mitochondria were labeled with MitoTracker Deep Red and loaded into the top chamber (Figure 3A). After incubation for 16 h, flow cytometry demonstrated that the MitoTracker Deep Red-labeled mitochondria penetrated through the HUVEC cell-coated membrane into the bottom chamber (Figure 3B), which was markedly increased in the presence of treatment with SDF-1α (Figure 3C). Flow cytometry showed the expression of CXCR4 on PB-derived mitochondria (Figure 3D). This was similar to the CXCR4 expression on the isolated mitochondria from peripheral blood platelets [12]. It suggests that the chemokine SDF-1α interaction with its receptor CXCR4 may be one of the major mechanisms contributing to the chemotaxis and homing of mitochondria. 

### 2.4. Immune Modulation of PB-derived Mitochondria

Our previous work established the immune modulation of platelet-derived mitochondria on human T cells [12]. To explore the immune modulation capabilities of PB-derived mitochondria, anti-CD3/CD28 bead-activated PBMC were treated with PB plasma-derived mitochondria at 200 µg/mL. After the treatment for 3 days, there were a number of bigger cell clusters at different sizes in the presence of PB-derived mitochondria group than those just activated with anti-CD3/CD28 beads (Figure 4A). Control PBMC failed to show a marked cluster formation (Figure 4A, left panel). Cell quantification confirmed that the proliferation of anti-CD3/CD28 bead-activated PBMC was markedly increased after the treatment with PB-derived mitochondria (Figure 4B, *P* = 0.034). Flow cytometry revealed that both percentages of activated CD4^+^HLA-DR^+^ and CD8^+^HLA-DR^+^ T cells were significantly increased in the presence of 200 µg/mL PB-derived mitochondria relative to only treatment with anti-CD3/CD28 beads (Figure 4C,D). Additionally, intracellular cytokine staining demonstrated that the percentages of IFN-γ^+^ cells, interleukin (IL)-12^+^ and IL-17A^+^ cells were markedly decreased following treatment with PB-derived mitochondria (Figure 4E,F and Figure 5F). There were no marked differences in the expressions of CD4 T helper type 2 (Th2)-associated cytokines IL-4 and IL-5, and other cytokines such as IL-1β, IL-10, IL-13, TNFα, and TGF-β1 (Figure 5A–H). These data suggest that PB-derived mitochondria display an immune modulation.

## 3. Discussion

The current data demonstrate the novel concept that mitochondria naturally circulating in the blood are present in human blood as well as multiple other animals. These findings suggest human plasma, FBS, and other related products should be used cautiously in any biological research that might be affected by mitochondrial contamination. Specifically, 10% concentration of FBS has been widely utilized for cell cultures and is now recognized to exhibit significant effects on cellular differentiation and functions in comparison with serum-free culture medium [16,17]. To overcome this issue and possible contamination with other extracellular vesicles in FBS [18], chemically defined serum-free culture media are preferred for biological research and translational studies [17]. To maintain and promote cell growth for clinical cell therapy, autologous plasma or serum may be applied to circumvent the allogenic mitochondria in the final clinical product during the good manufacturing practice (GMP) production. 

Since RBC and platelets comprise 99% of the cellular component of blood [19] and there are no mitochondria in mature RBC [15], it is likely that these circulating mitochondria are primarily released by platelets [12]. Our previous work showed that platelets and platelet-derived mitochondria act as novel immune modulators to induce immune tolerance through the expressions of PD-L1 and CD270 [12], which bind to their ligands programmed death receptor-1 (PD-1) [20,21] and BTLA (B and T lymphocyte attenuator) [22]. Similarly, both CB- and PB-plasma-derived mitochondria expressed PD-L1 and CD270. To this end, the percentage of Th1 cytokine (IFN-γ)- producing cells was down-regulated after the treatment with PB plasma-derived plasma. IFN-γ is a pleiotropic cytokine and predominantly produced by CD4 T helper type 1 (Th1) cells and CD8 cytotoxic T lymphocytes (CTL) effector T cells, and natural killer (NK) cells [23,24]. Substantial evidence demonstrated that a key player in governing the differentiation of Th1 cells is IL-12, a cytokine mainly produced by antigen-presenting cells (APC) such as dendritic cells (DC) [25,26]. Therefore, current data suggest that plasma mitochondria exhibited the suppressive effects on Th1 cells, potentially associated with their expressions of immune tolerance markers CD270 and CD274 (PD-L1) on plasma mitochondria. By comparison with directly-isolated mitochondria from platelets [12], PB plasma-derived mitochondria improved the proliferation of anti-CD3/CD28 bead-activated PBMC, probably due to the difference of mitochondrial contents with some potential growth factors or the mixed with other extracellular vesicles (EVs) in plasma mitochondria. Additionally, PB-derived mitochondria might be phagocytosed by monocytpes and/or dendritic cells in PBMC and caused their activation, leading to the stimulation of PBMC proliferation and T-cell activation. 

Recently, we demonstrated that platelet-derived mitochondria express chemokine receptor, leading to the migration of mitochondria to pancreatic islets through transwell membranes and subsequent enhancement of human islet β-cell functions [12]. Possibly because of the expression of embryonic stem (ES) cell markers in platelet-derived mitochondria [12], ongoing studies have demonstrated that treatment with the purified platelet-derived mitochondria can reprogram the transformation of adult peripheral blood insulin-producing cells (PB-IPC) [27] into pluripotent stem cells, leading to the generation of neuronal cells and functional CD34^+^ hematopoietic stem cells (HSC)-like cells that could produce blood cells such as T cells, monocytes/macrophages, granulocytes, red blood cells, and megakaryocytes (MK)/platelets. These findings reveal a novel function of mitochondria directly contributing to cellular reprogramming. Due to their small sizes (50–400 nm), circulating mitochondria may easily pass through tissue barriers (e.g., the blood-brain barrier) via chemotaxis as demonstrated by the current study, contributing to the regeneration of aged or damaged tissues via cell reprogramming and potential correction of chronic inflammation as observed in Alzheimer’s disease. Thus, the circulating mitochondria may function as a novel mediator, leading to the energy balance and cross-talk among cells, tissues, and organs, and maintenance of homeostasis.

## 4. Materials and Methods 

### 4.1. Isolation of Mitochondria from Human Blood Plasma

To examine mitochondria in adult human blood, plasma samples were isolated from adult volunteer donors and purchased at New York Blood Centers. To examine the mitochondria in human cord blood, human umbilical cord blood units (50–100 mL/unit) were collected from healthy donors and purchased from Cryo-Cell International blood bank (Oldsmar, FL, USA). Cryo-Cell has received all accreditations for cord blood collections and distributions, with hospital IRB approval and signed Consent Forms from donors. Initially, all plasma samples were centrifuged at 3000 rpm × 15 min to remove all cellular components (including platelets), and followed by filtered with 0.45 μm(Argos Technologies, Vernon Hills, IL) syringe filters to remove all cellular debris or bigger vesicles in the plasma (Figure 1A). Consequently, the filtered plasma samples were centrifuged at 12,000 *g* × 15 min at 4 °C. The pellets of mitochondria were collected and tested with different mitochondrial markers by flow cytometry. For transmission electronic microscope, plasma samples were collected by two-step centrifuges, without filtering. The concentration of mitochondria was determined by the measurement of protein concentration using a NanoDrop 2000 Spectrophotometer (ThermoFisher Scientific, Waltham, MA, USA). The mitochondria isolated from platelets served as positive control, as previously described [12]. Briefly, the mitochondria were isolated from apheresis platelets (New York Blood Center) using the Qproteome Mitochondria Isolation kit (Qiagen, Hidden, Germany) according to the manufacturer’s recommended protocol. 

### 4.2. Electron Microscopy and Isolation of Mitochondria from Animal’s Blood

For electron microscopy, the protocol for the isolation of mitochondria was performed as described above, except without a filtration with 0.45 µm filter. Samples were fixed in 2.5% gluteraldehyde/4% paraformaldehyde in 0.1 M Cacodylate buffer. Samples were then post-fixed in buffered 1% Osmium Tetroxide, dehydrated in a graded series of acetone and embedded in epon resin. The 90 nm-thin sections were cut on a Leica EM UC6 ultramicrotome. The sectioned grids were stained with saturated solution of uranyl acetate and lead citrate. Images were captured with an AMT XR111 digital camera on a Philips CM12 transmission electron microscope.

To determine the mitochondria in animal’s blood, fetal bovine serum (FBS) were purchased from different vendors such as Sigma-Aldrich (St. Louis, MO, USA), Atlanta Biologicals (Premium Select, Catalog # S11550, Flowery Branch, CA, USA), Gibco (ThermoFisher Scientific, Catalog # A3160402, Rockville, MD, USA), HyClone (Charcoal/Dextran Treated, Catalog #SH30068.01, Swedesboro, NJ, USA) and Clontech (Tet System Approved FBS, Catalog # 631106, Mountain View, CA, USA), along with heat-inactivated horse serum (Life Technologies, Carlsbad, CA, USA). 

For mitochondrial staining with fluorescent dyes, plasmas were labeled with MitoTracker Deep Red FM (100 nM) (Catalog number: M22426, Thermo Fisher Scientific, Waltham, MA, USA) at 37 °C for 15 min according to the manufacturer’s recommended protocol, followed by two washes with PBS at 12, 000 rpm 15 min at 4 °C.

### 4.3. PBMC Isolation and Proliferation Assay

Human buffy coat blood units were purchased from the New York Blood Center (New York, NY). Human peripheral blood-derived mononuclear cells (PBMC, *N* = 4 biological repeat) were harvested as previously described [13]. PBMC (5 × 10^5^ cells/well) were stimulated with Dynabeads coupled with anti-CD3 and anti-CD28 antibodies (Life Technologies, Waltham, MA) for 72 hours in the presence of treatment with PB-derived mitochondria at 200 µg/mL in duplicate respectively and were incubated at 37 °C in 5% CO_2_ in the tissue culture-treated 96-well plate. Mitochondria-untreated cells served as control. After the treatment for 3 days, cells were collected for cell count with trypan blue staining and assesses cell viability via trypan blue exclusion by using TC20™ Automated Cell Counter (Bio-Rad, Hercules, CA, USA). Consequently, cell samples were prepared for flow cytometry [12]. 

### 4.4. Flow Cytometric Analysis

Flow cytometric analyses of surface and intracellular markers were performed as previously reported [12]. The MitoTracker Deep Red-labeled mitochondria were washed with PBS at 12,000 *g* for 15 min at 4 °C. Samples were pre-incubated with human BD Fc Block (BD Pharmingen, NJ, USA) for 15 min at room temperature, and then directly aliquoted for different antibody staining. For surface staining, mitochondria were stained for 30 min at room temperature with the immune tolerance-related markers anti-human CD270 mAb (BioLegend, San Diago, CA, USA) and CD274 mAb (Invitrogen, Waltham, MA, USA), PE-conjugated anti-human CXCR4 (Beckman Coulter, Brea, CA, USA), and then washed with PBS at 12,000 *g* for 15 min prior to flow analysis. Isotype-matched mouse anti-human IgG antibodies (Beckman Coulter, Brea, CA, USA) served as negative controls. For intra-mitochondria staining, MitoTracker Deep Red-labeled mitochondria were fixed and permeabilized by using the PerFix-nc kit (Beckman Coulter, Brea, CA, USA) according to the manufacturer’s recommended protocol, followed by the staining with FITC-conjugated anti-human heat shock protein 60 (Hsp60) mAb (BD Biosciences, La Jolla, CA, USA). Isotype-matched IgGs served as control. 

To characterize phenotypic changes, the mitochondria-treated and untreated PBMC were performed by flow cytometry with associated markers including Pacific Blue (PB)-conjugated anti-human CD3 mAb (BioLegend, San Diago, CA, USA), FITC-conjugated anti-human CD4 mAb (Beckman Coulter, Brea, CA, USA), APC-Alexa Fluor 750-conjugated anti-CD8 mAb (Beckman Coulter, Brea, CA, USA), and BV 510-conjugated anti-human HLA-DR mAb (BD Pharmingen, NJ, USA). Isotype-matched IgGs served as control. 

For intracellular cytokine staining, mitochondria-treated and untreated PBMC were stained with following antibodies including FITC-conjugated anti-human IFN-γ mAb (Beckman Coulter, Brea, CA, USA), PE-conjugated anti-human IL-12 mAb, PB-conjugated anti-human TNF-α mAb, PE-CY7-conjugated anti-human IL-4 mAb, PE-conjugated anti-human IL-5 mAb, FITC-conjugated anti-human TGF-β1 mAb, PB-conjugated anti-human IL-1β mAb, PE-CY7-conjugated anti-human IL-13 mAb, APC-conjugated anti-human IL17A and IL-10 mAbs (BioLegend, San Diago, CA, USA). Cells were analyzed using a Gallios Flow Cytometer (Beckman Coulter, Brea, CA, USA) equipped with three lasers (488 nm blue, 638 red, and 405 violet lasers) for the concurrent reading of up to 10 colors. The final data were analyzed using the Kaluza Flow Cytometry Analysis Software version 2.1 (Beckman Coulter, Brea, CA, USA).

### 4.5. Transwell Experiment and Migration Assay of Mitochondria 

To investigate the migration of mitochondria and whether they can pass through the endothelial cells, the transwell system (Corning, Life Sciences, France) with 0.45 μm porous size was initially coated with human umbilical vein endothelial cells (HUVECs) at 1 × 10^5^ cells/mL, 1 mL/well in complement endothelial cell basal (EBM-2) medium (Lonza, Walkersville, MD, USA) and incubated for 24 h at 37 °C in 5% CO_2_, and followed by adding the MitoTracker Deep Red FM-labeled plasma mitochondria to the top chamber of transwell system, in the presence or absence of stromal cell-derived factor 1α ( SDF-1α, Tonbo Biosciences, CA, USA) at 40 ng/mL in triplicate in the bottom chamber respectively. After the treatment for 16 h at 37 °C 5% CO_2_ culture conditions, the medium from the bottom chamber was collected to examine the migration of mitochondria by flow cytometry. 

### 4.6. Quantitative Real Time PCR Array

Expression of different mRNAs was analyzed by quantitative real-time PCR [12]. Total RNA was extracted from the isolated mitochondria using a Qiagen kit (Qiagen Inc., Valencia, CA, USA). First-strand cDNAs were synthesized from total RNA using an iScript gDNA Clear cDNA Synthesis Kit according to the manufacturer’s instructions (Bio-Rad, Hercules, CA, USA). Real-time PCR was performed using the StepOnePlus Real-Time PCR System (Applied Biosystems, CA, USA) under the following conditions: 95 °C for 10 min, then 40 cycles of 95°C for 15 s, and 60 °C for 60 s. For RT2 Profiler real time PCR Array, the RT2 ProfilerTM PCR Array Human T-Cell & B-cell Activation kit (96-well format, PAHS-053Z, Qiagen) was used according to the manufacturer’s instructions. The data were analyzed using PrimePCR array analysis software (Bio-Rad). 

### 4.7. Statistical Analysis

All experiments were performed at least three times, and statistical analysis was carried out using Graphpad Prism 8 (version 8.0.1) software. The normality test of samples was performed by the Shapiro-Wilk test. Statistical analyses of data were performed by the two-tailed Student’s *t*-test to determine statistical significance for parametric data. Values were given as mean SD (standard deviation). Statistical significance was defined as *P* < 0.05, with two-sided.

## Figures and Tables

**Figure 1 ijms-21-02122-f001:**
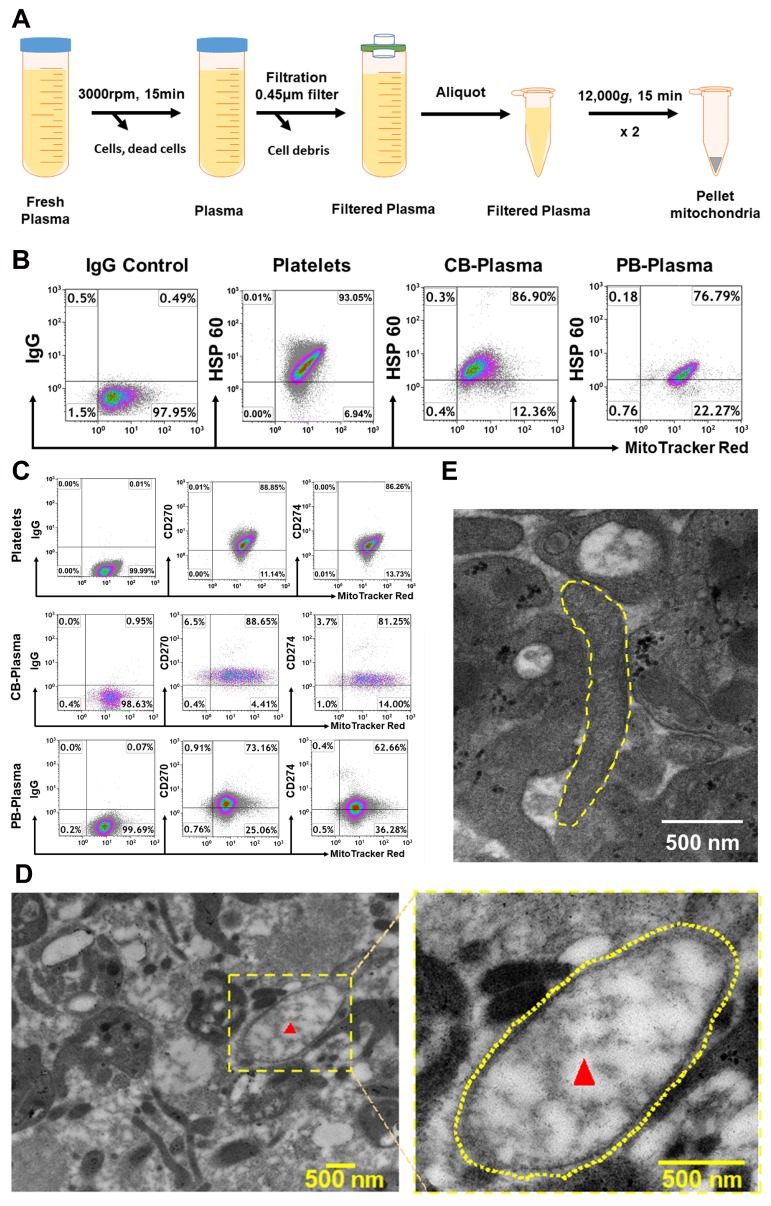
Characterization of mitochondria in human blood. (**A**) Outline the protocol for an isolation of mitochondria from plasma. (**B**) Flow cytometry data show the plasma-derived mitochondria positive for specific mitochondrial markers (MitoTracker Deep Red) and heat shock protein (HSP) 60 from the plasma of human cord blood (CB, *N* = 4) and adult peripheral blood (PB, *N* = 4). Mitochondria isolated from the plasma of peripheral blood (*N* = 4) served as positive control. (**C**) Expression of immune tolerance-associated markers CD270 and CD274 on human CB (*N* = 8) and PB (*N* = 9) plasma-derived mitochondria respectively. Mitochondria isolated from peripheral blood-derived platelets served as positive control. Isotype-matched IgGs served as negative control. (**D**) Electron microscopy demonstrating the free mitochondria (red arrow, left) in the plasma of adult human blood, with a high magnification (yellow dashed circle, right). (**E**) Electron microscopy show a free elongated mitochondrion (yellow dashed circle) in adult human blood plasma.

**Figure 2 ijms-21-02122-f002:**
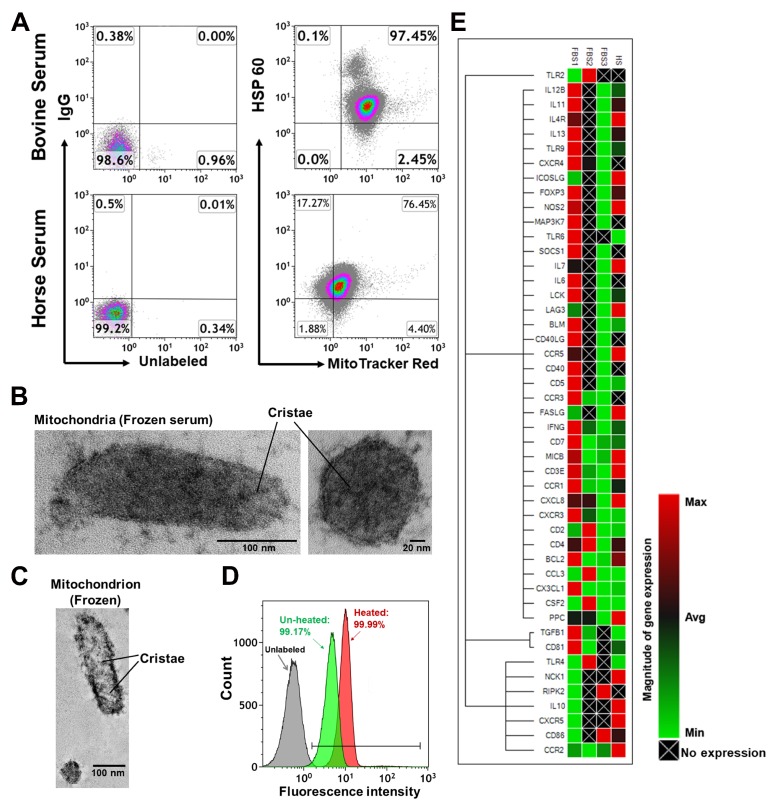
Examination of mitochondria in animal blood. (**A**) Flow cytometry data demonstrating mitochondria in fetal bovine serum (FBS, *N* = 4) and horse serum (HS, *N* = 3). Isotype-matched IgGs served as control. (**B**) Electron microscopy demonstrating mitochondria in the commercialized frozen FBS. (**C**) Electron microscopy demonstrating mitochondria in the commercialized frozen FBS. (**D**) Detection of mitochondria in FBS after treatment (red) or no treatment (green) with heat inactivation. Mitochondria unlabeled with MitoTracker Deep Red served as the control (grey). Representative data were from one of four experiments. (**E**) Expression of human T- and B-cell activation–associated genes by real time PCR array. Mitochondria were isolated from FBS and horse serum, followed by real-time PCR array analysis with a kit for human T and B cell activation markers.

**Figure 3 ijms-21-02122-f003:**
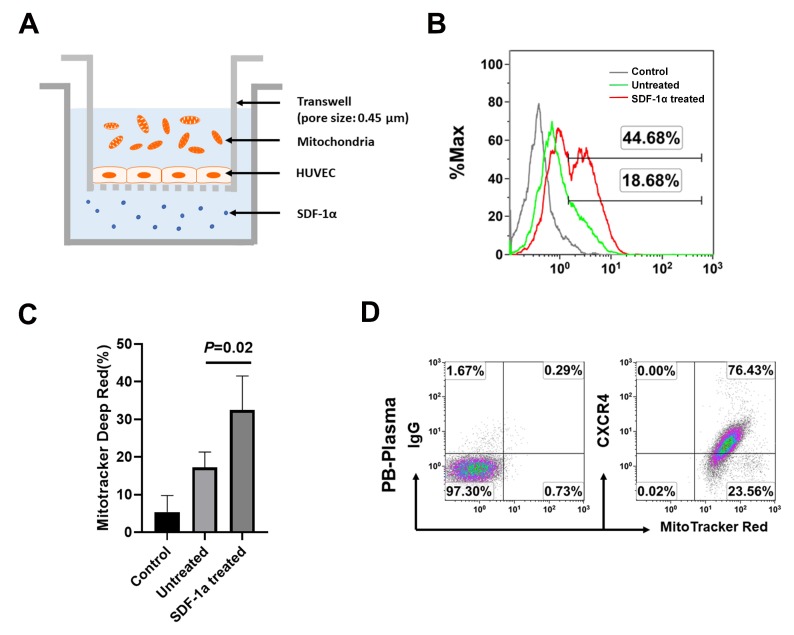
Chemotaxis of plasma mitochondria by transwell migration assay. (**A**) Experimental design of transwell migration assay. Human umbilical vein endothelial cells (HUVEC) were seeded in the top chamber at 1 × 10^5^ cells/mL. The chemoattractant SDF-1α was added into the bottom chamber. (**B**) Flow cytometry showed the migration of mitochondria into the bottom chamber. After incubation for 16 h, flow cytometry demonstrated the migration of MitoTracker Deep Red-labeled mitochondria was increased in the presence of treatment with SDF-1α (red histogram), relative to the absence of SDF-1α (green histogram). Unlabeled mitochondria served as negative control (grey histogram). The data were representative of three experiments with similar results. (**C**) Improve the migration of mitochondria after the treatment with SDF-1α (*N* = 6). (**D**) Flow cytometry showed the expression of CXCR4 on PB-derived mitochondria (*N* = 4). Isotype-matched IgG served as control.

**Figure 4 ijms-21-02122-f004:**
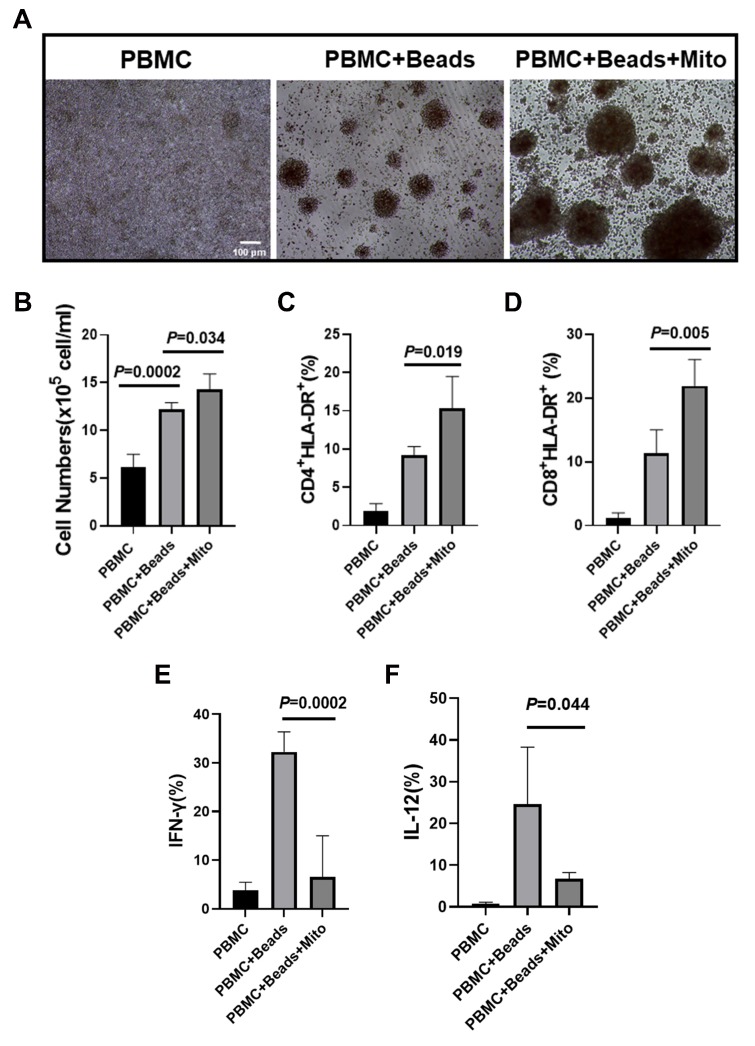
Immune modulation of peripheral blood (PB)-derived mitochondria on PBMC. Health donor-derived PBMC (*N* = 4) were treated with PB-derived mitochondria (*N* = 4) in duplicate for 3 days in the presence of T-cell activator CD3/CD28 Dynabeads. (**A**) Phase contrast image show the cluster formation with different sizes. Untreated PBMC served as control (left panel). (**B**) Increase the PBMC proliferation by PB-derived mitochondria. (**C**) Improve the percentage of the activated CD4^+^ HLA-DR^+^ T cells by PB-derived mitochondria. (**D**) Improve the percentage of the activated CD8^+^ HLA-DR^+^ T cells by PB-derived mitochondria. (**E**) Suppression of inflammatory cytokine IFN-γ production in the presence of PB-derived mitochondria at 200 µg/mL. (**F**) Decrease the level of cytokine IL-12 in CD3/CD28 Dynabead-activated PBMC after the treatment with PB-derived mitochondria at 200 µg/mL. Data were given as mean ± SD (standard deviation, *N* = 4).

**Figure 5 ijms-21-02122-f005:**
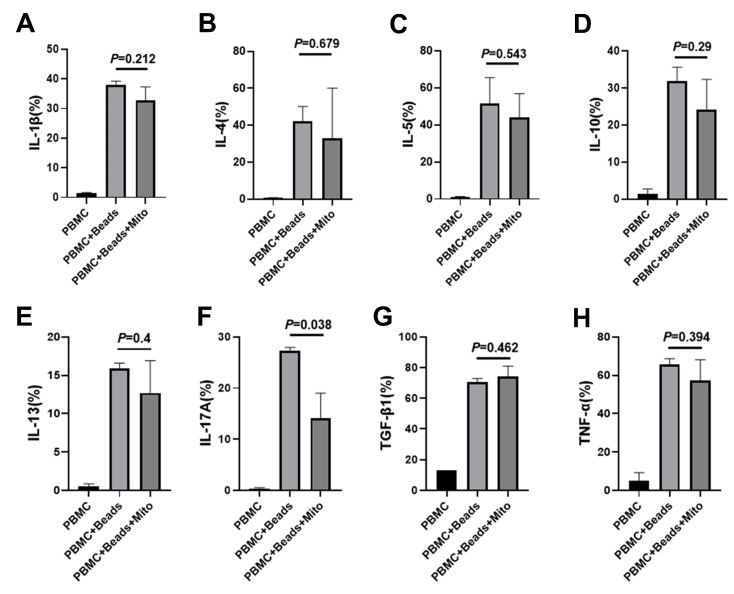
Intracellular cytokine analysis by flow cytometry. Health donor-derived PBMC (*N* = 4) were treated with PB-derived mitochondria at 200 µg/mL (*N* = 4) in duplicate for 3 days in the presence of T-cell activator CD3/CD28 Dynabeads. (**A**–**H**) There were no marked differences in the percentages of IL-1β (**A**), IL-4 (**B**), IL-5 (**C**), IL-10 (**D**), IL-13 (**E**), TGF-β1 (**G**), and TNF-α (**H**)-producing cells after the treatment with PB-derived mitochondria. (**F**) Downregulation of the percentage of IL-17A-positive cells after the treatment with PB-derived mitochondria. Data were given as mean ± SD (standard deviation, *N* = 4).

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
