# Peer review of "Existence of Circulating Mitochondria in Human and Animal Peripheral Blood"

_ijms, 2020, doi:10.3390/ijms21062122_

Round 1

Reviewer 1 Report

Excellent results and well-designed study with a high profile of findings.

Author Response

Many thanks for your kind consideration!

Yong

Reviewer 2 Report

This is an interesting study and overall well designed and carried out, but some aspects should be clarified:

  • What was the yield of isolated mitochondria from the different sources? (i.e number of mitochondria / mL sample).
  • Have the isolated mitochondria from the different sources the same organelle activity? (i.e characterization of membrane potential, cyto c content, …).
  • Have the isolated mitochondria from the different sources the same immunomodulatory potential? (i.e mitochondria dose- cellular response in-vitro assays comparing the different sources, …).
  • What are the directly affected immune populations by isolated free mitochondria?. In-vitro assays might be conduced on sorted immune cell populations, in order to establish the right conclusions. Indeed, as found in the real-time PCR array, B cells could be also targets.
  • Are the in-vitro activation of T cells directly mediated by CD270 and PD-L1 expression on the added mitochondria? (i.e. can be the effects abrogated by anti-CD270 or/and anti-CD274?).
  • The results from flow cytometry might include mean and standard deviation data in the relevant gates.

Some minor revisions and unclear questions might be answered or corrected:

  • When speaking about the source of mitochondria, nomenclature (blood or plasma) must be precise in order not to confuse blood and plasma concepts (Figure 1: “…protocol for isolation of mitochondria from blood”, while the figure A indicates “Fresh plasma”; Materials and Methods, line 216: “Initially, all plasma samples…”
  • In some parts of the text, greek symbols such as microns or gamma are lost. 
  • Figure 5, some specification needs to be added: “There were no marked differences in the percentages of […] producing cells after the treatment with…”

- Materials and Methods:

4.1 section: mitochondria from platelets are used as a positive control, but there is no reference or description of the isolation method developed for them.

4.3 section: some data or methods must be included. How many PBMC were stimulated?. How the cell count showed in Figure 4B was carried out?.

4.4 section: Where was MitoTracker Deep Red purchased from?.

- References: references 12 and 16 are the same.

- Further studies: as mentioned by the authors, some aspects (as indicated below) need to be clarified.

  • Does determine the health state the releasing profile (rate, immunomodulating activities) of free mitochondria? (i.e. healthy vs patients affected by immunosuppression or autoimmunity?).
  • What is/are the cell/tissue source/s for the free mitochondria found circulating? (i.e analysis of lymphatic fluid).
  • Regarding the mechanisms for the immunomodulatory activities: are they only dependent on cellular contact (by expression of receptors or ligands as CD270 and PD-L1) or also free circulating mitochondria are able to produce soluble mediators?.

Author Response

Please see the attached file for responses to your questions. Many thanks. 

Yong

Reviewer 3 Report

In this manuscript, Song et al describe the existence and the potential effects of circulating mitochondria on the regulation of T cell responses. The authors characterized the circulating mitochondria and analyzed the function of them. Most importantly, the circulating mitochondria were also found in animal serum products which are routinely be add in culture medium as supplements. The findings are potentially interest and important, however, there are several points need to be explained or discussed.

  1. For the characterization of mitochondria in CB or PB, why choose CD270 and CD274. Are there any other immune modulatory molecules express on MT? Also, this molecules are supposedly expressed on cell membrane, what are the possible mechanism that guide these molecules to MT.
  2. For the figure 2E and description in the results line 90-92, it is confusing that using human T and B activation real-time PCR array kit to analyze mRNA profile of MT from FBS and horse serum. Also, it is unlikely that those genes listed in the figure are encoded by MT genome, please explain this.
  3. In figure 4, it seems that PB-derived MT promotes T cell activation (Fig. 4A-E), however, MT also express PD-L1 which may suppress T cell activation. In the following results (Fig4E and F), PB-MT inhibits IFN and IL-12 production. The data presented here may have conflict and need to be explained.
  4. In figure 4 and 5, what are the population gated for analyze the percentage of IFN and other cytokine production, CD4 or CD8? Also, in figure 5, most proportion of cells expressed IL-4, IL-17A and others upon activation, please check this.
  5. In discussion part, line 168, although the presence of MT in FBS which may affect cell fate during the culture, it is important to know the effective concentration of MT in the culture media may alter the function of the cells. In the functional study in figure 4, it may need to titrate the concentration of MT before the statement be made in discussion part.
  6. In line 194, platelet-derived mitochondria express “chemokines”, leading to the migration of mitochondria to pancreatic… please check “chemokines” or “chemokine receptors”
  7. Line 299, Qiangen or Qiagen?

Author Response

(The authors gave the same response as above.)
